# Hydrolyzed Rice Protein-Based Formulas, a Vegetal Alternative in Cow’s Milk Allergy

**DOI:** 10.3390/nu12092654

**Published:** 2020-08-31

**Authors:** Christophe Dupont, Alain Bocquet, Daniel Tomé, Marie Bernard, Florence Campeotto, Pascale Dumond, Anna Essex, Marie-Laure Frelut, Lydie Guénard-Bilbault, Gideon Lack, Agnès Linglart, François Payot, Alain Taieb, Nicolas Kalach

**Affiliations:** 1Pediatric Gastroenterology Department, Paris-Descartes University, 75006 Paris, France; 2Pediatric Gastroenterology Department, APHP Necker-Enfants Malades Hospital, 750015 Paris, France; florence.campeotto@aphp.fr; 3Marcel Sembat Clinic, 92100 Boulogne, France; 4Association Française de Pédiatrie Ambulatoire (French Association of Ambulatory Paediatrics), 33400 Talence, France; bocquet.a@wanadoo.fr (A.B.); frelut@club-internet.fr (M.-L.F.); 5Medical and Pharmacy School, Franche-Comté University, 25000 Besançon, France; 6UMR Nutrition Physiology and Ingestive Behavior, AgroParisTech, INRA, Paris-Saclay University, 75005 Paris, France; tome@agroparistech.fr; 7Charles Nicolle Hospital, 76031 Rouen, France; marieb1997@gmail.com; 8Pediatric Allergy Department, Children’s Hospital, University Hospital of Nancy, 54000 Vandoeuvre les Nancy, France; p.dumond@chru-nancy.fr; 9Sodilac Employee, Sodilac, 92000 Levallois, France; anna.essex@sodilac.com; 10Pediatric Practice, 16 Rue de Sept Fonds, 81000 Albi, France; 11Medical General and Allergy Practice, 67400 Illkirch Graffenstaden, France; guenard-bilbault@orange.fr; 12Paediatric Allergy Research Group, Department of Women and Children’s Health, School of Life Course Sciences, King’s College London, London WC2R ELS, UK; gideon.lack@kcl.ac.uk; 13INSERM-U1185, Paris Sud Paris-Saclay University, 75005 Paris, France; agnes.linglart@aphp.fr; 14APHP, Reference Center for Rare Disorders of the Calcium and Phosphate Metabolism, Network OSCAR and ‘Platform of Expertise Paris Sud for Rare Diseases, 75005 Paris, France; 15APHP, Endocrinology and Diabetes for Children, 75005 Paris, France; 16Pediatric Practice, 69006 Lyon, France; francoispayot@wanadoo.fr; 17Department of Dermatology and Pediatric Dermatology, Bordeaux University Hospital, 33000 Bordeaux, France; alain.taieb@u-bordeaux.fr; 18Department of Paediatrics, Saint Antoine Paediatric Hospital, Saint Vincent de Paul Hospital, Groupement des Hôpitaux de l’Institut Catholique de Lille (GHICL), Catholic University of Lille, 59000 Lille, France; Kalach.Nicolas@ghicl.net

**Keywords:** children, rice, hydrolyzed protein, cow’s milk allergy

## Abstract

Formulas adapted to infant feeding, although most of the time made from cow’s milk proteins, can be made from hydrolyzed rice protein but they must be classified as “formulas for specific medical needs”, according to European regulations. The nutritional quality of rice proteins is thus suitable to be used in infant formulas giving that it is supplemented by certain amino acids which can be lacking. Besides, hydrolysis is required to facilitate their water solubility and digestibility. Owing to a low allergenicity of rice and to the absence of the cross-allergy between milk proteins and rice proteins, these formulas are adapted to the diet of children with cow’s milk protein allergy (CMPA), which explains their growing use in some countries. However, CMPA, an expanding disorder, has consequences for growth, bone mineralization, and often has an association with allergy to other foods, including cow’s milk extensive hydrolysate, so that a surveillance of the adaption of hydrolyzed rice protein formulas (HRPF) to CMPA, the absence of unexpected side effects, and the appropriate response to its various health hazards seems mandatory. This paper analyses the health problem deriving from CMPA, the industrial development of hydrolyzed rice protein formulas, and the limited number of clinical studies, which confirms, at the moment, a good allergic tolerance and safety. The goal is to better advise heath care professionals on their use of HRPFs during CMPA.

## 1. Introduction

Cow’s milk protein allergy (CMPA) is the most common food allergy in childhood and its prevalence ranges between 1.9% and 4.9% [1]. Most children with CMPA are less than one year old and may require a hypoallergenic formula in the absence of breast milk. As recently reviewed [2], in most countries, hypoallergenic formulas include both extensively hydrolyzed formula (CMP-eHF) and amino acid formula (AAF).

In the last two decades, formulas based on hydrolyzed rice protein (HRP) have been developed and they seem to be a valid alternative to cow’s milk protein (CMP) formulas, both as a vegetal alternative for families interested in plant-based feeding and for children with CMPA [3]. HRPFs may be considered as a part of a trend towards plant-based sources of protein, which may be beneficial on human health, mortality [4] and may have an impact on the environment, hence its potential growing use in infants and children.

This paper aims towards a better understanding of HRP formulas (HRPF), of the characteristics of rice proteins, their appropriateness to feeding CMPA children, their suitability to restore normal growth, and their potential interaction with the acquisition of CMP tolerance. The paper thus first describes CMPA characteristics and its nutritional hazards, overviews plant protein in infant nutrition and the manufacturing of HRPFs, explains the European legal framework and the market availability of such formulas, and finishes with the current knowledge on HRPF’s efficiency in the treatment of CMPA.

## 2. Cow’s Milk Protein Allergy (CMPA) and Its Nutritional Hazards

CMPA is an abnormal body immune system response to cow’s milk proteins and one of the most common food allergies in childhood. In infants with CMPA, cow’s milk cannot be replaced by milk from sheep, goats, or buffalo due to a high cross reactivity between milk proteins from mammals. Symptoms of milk allergy range from mild to severe, and may be digestive, dermatological, or respiratory. The immune mechanisms involved during CMPA may be IgE-mediated or non-IgE mediated or mixed, particularly in young infants. Among clinical symptoms, nutritional troubles are frequent, involving growth faltering, stagnation, and failure, a deficit in macronutrients and micronutrients and feeding [2].

For all these reasons, guideline publications on food allergy (FA) highlight the importance of nutritional advice by a specialized dietitian/nutritionist [5]. For the same reasons, the above-mentioned issues need to be addressed if HRPFs are to be used during CMPA.

### 2.1. Height and Weight Problem

CMPA may affect weight, height, or BMI (i.e., a reduced mean (SD) height for age and weight for age) as suggested by studies already in the 1990s [6], a phenomenon that is currently widely investigated [5]. Growth may be affected by CMPA prior to the diagnosis of CMPA and the beginning of treatment, and may persist during the CMP elimination regimen, with a frequent delay in diagnosing CMPA increasing the risk of undernutrition [7,8].

The pathophysiology of growth impairment during CMPA involves inflammatory processes, e.g., increased production of cytokines, increased intestinal permeability, increased metabolic rate, GH resistance, and reduced appetite. In addition, comorbidities play a role, like atopic dermatitis, eosinophilic esophagitis, and aversive feeding behavior. Finally, the elimination diet likely slows the growth recovery because of unpalatability, reduced caloric intake, and reduced specific nutrient intake [9,10].

Several studies report the role of an association of CMPA with multiple food allergies, and of other comorbid factors, among which is an inappropriate diet [11,12,13]. A spectrum of growth and weight gain anomalies can be drawn, from severe growth retardation in children with CMPA, multiple food allergies and other risk factors, to a mild growth impact in children with CMPA, a controlled diet and adequate nutrients, and a lack of catch up growth in healed children, and having a standard diet.

### 2.2. Micronutrients and Macronutrients Deficiency

CMPA, especially during multiple food allergy, may put the child at risk of macro- and micronutrient deficiencies. The role of FA itself in the genesis of these deficiencies remains unclear. Studies have shown that children with a multiple elimination diet had a lower intake of some key micronutrients including vitamins A, D, E, C; folic acid; calcium; zinc; iron; and B vitamins [14]. Maslin et al., comparing 28 CMPA children and 73 non-allergic controls, found significant deficiency in micronutrients, i.e., riboflavin, iodine, sodium, and selenium, but without any reduction in intake [15].

A deficiency in vitamin D, calcium, and phosphorus, essential for bone growth and metabolism, may reduce bone mineralization [2], especially during CMPA, since milk and dairy products are an important source of calcium and phosphorus. However, the mechanisms involved seem complex. These have been reviewed recently [8].

The effect of bone mineral density (BMD) on CMPA was studied in small populations of pre-pubertal children and of young adults [16]. In lumbar spine, z-scores were lower in CMPA children (*n* = 52) than in controls [16]. Calcium intake respected the American Recommended Daily Allowance (RDA) in 39% of CMPA children vs. 74% of controls (*p* = 0.003). Calcium intake remained below two-thirds of RDA in more than 20% of CMPA children. A dairy-free diet longer than one year puts the prepubertal children with CMPA at risk of increased bone resorption, as suggested by the increase in receptor activator of nuclear factor kB ligand (RANKL) level and the decrease in the the osteoprotegerin/RANKL ratio [17]. In food consumption surveys, milk and other dairy products amount to 45% of the total daily intake of calcium in 2–18-year-old children from the United States, to 47% in those from Australia, and to 49% in 3–17-year-old children from France [18]. In infants less than 6-months old, the contribution to total daily calcium intake of human milk and of infant formulas amounts to 90%, and, following complementary feeding, remained at 60% in 6- to 12-month-old children in a US survey [19]. More CMPA children than controls had calcium consumption below the recommended amount, 15 of 26 (57.6%) vs. 12 of 39 (30.7%) (*p* < 0.05). In contrast, feeding with an adapted infant/toddler formula or a calcium-fortified soy beverage allowed 91% (10/11) of them to meet the dietary reference intakes for calcium and vitamin D [14]. A decreased intake of calcium might decrease bone remodeling more (decrease bone formation) than an increase in bone resorption [20]. During CMPA, the intake of many nutrients (energy, proteins, potassium, zinc [21]) is reduced and may prevent reaching the peak bone mass, and thus impact on bone health.

### 2.3. Feeding Difficulties

Feeding difficulties are a common parental complaint in food allergic children, particularly in those with CMPA under elimination diet. Numerous wordings are used such as food refusal, fussy eating, aversive feeding, etc. Maslin et al. [22] reported a population of 101 children, 28 CMPA and 73 non-allergic controls, with a significant difference in allergic children, with altered taste preferences. The same authors also showed that children with CMPA submitted since a young age and for a long period of an elimination diet will develop persistent food difficulties (slow eating attitude, tastes disturbance food refusal, fussy eating, etc.) as compared non-CMPA children [22]. Those feeding difficulties are commonly reported in the literature in 40% of children with non-IgE-mediated CMPA and 16% of children with eosinophilic esophagitis (EoE) [23].

### 2.4. The Need for a Precise Dietary Follow-Up

A longitudinal follow-up of daily intakes is crucial, ideally with the help of a professional dietician. Feeding a child with enough of a formula adapted to feeding infants and children with CMPA remains the best option. Nutrition counseling proved to significantly improve nutrient intakes. Among children with FAs, the number of children meeting 67% of the Dietary Reference Intake for calcium and vitamin D was higher in families with nutrition counseling [14]. In Italy, the latter restored the total energy intake of 85 children with FA to values of control children and improved anthropometric and laboratory biomarkers of nutritional status [21]. The extent of the food elimination diet had no impact on growth or nutritional status of children with FA, when the diet was adequately supplemented [24]. In the largest American retrospective study, cow’s milk avoidance was deleterious on weight and height only in children aged 2–5 years, suggesting that parents may not substitute cow’s milk as well between 0 and 2 years [25]. Since solids become a larger portion of the diet in older children, parents/caregivers may not feel that milk substitution is important. Moreover, the elimination of typical snack foods, frequently consumed by preschoolers and mostly containing cow’s milk, leads to lower energy, as well as to lower protein and other nutrient intakes. For vitamin D, a blood assay may detect insufficient levels, but supplementing the child throughout the elimination diet is common practice. The vitamin D status was significantly lower in Finnish schoolchildren with a history of CMPA, indicating that restricting the diet may have long-term consequences on dietary habits and subsequently on later vitamin D status [26]. Calcium supplements may be used if intakes remain below the daily need. Follow-ups should be offered even after the child has outgrown CMPA to encourage progression to an unrestricted diet and to prevent eating disorders.

## 3. Plant Proteins in Infant Nutrition

The protein content varies according to plants, in quantity and quality. The protein content of rice is qualitatively pretty well adapted to human physiology, considered poorly allergenic, and present in amounts allowing its use in industrial processes.

### 3.1. Protein and Amino Acid Requirements and Reference Patterns for Infants

Dietary proteins provide nitrogen and amino acids that are required for the synthesis of proteins, of other nitrogenous compounds, and for various metabolic pathways, being precursors of metabolically active compounds, such as neurotransmitters, glutathion, haem, and creatine.

The quantity and the amino acid profile are both limiting factors of the quality of protein supply. In both adults and children, there is a need for indispensable amino acids (IAA), which need to be provided preformed into the diet [27] to support physiological functions and growth. Several methods may assess the quality of protein in a diet or food [28], but the currently accepted and preferred approaches are the Scoring methods such as the Protein Digestibility-Corrected Amino Acid Score (PDCAAS) and the Digestible Indispensable Amino Acid Score (DIAAS) that relate the IAA content of an individual foodstuff or mixed diet to a reference IAA profile calculated to meet IAA needs and specific for each age range, and corrected for protein or IAA digestibility [29,30,31].

Reference IAA profiles have been proposed according to ages [27]. For the first six months of life, when infants are theoretically only breastfed, the composition of breast milk and the amounts ingested provide relatively robust estimates of energy and amino-acid requirements. For children aged 6 to 36 months, protein and indispensable amino acid requirements are calculated by a factorial approach, i.e., the addition of requirements for both maintenance and tissue deposition during growth. It is assumed that requirements for maintenance are identical to adult values and that growth requirements derive from available data on protein accretion and from the amino acid composition of body proteins, adjusted for efficiency of dietary protein utilization (0.58) [27,32]. The amino acid reference patterns are then estimated from human milk composition or calculated based on the evaluation by a factorial approach of protein and amino acid requirement for children 1 to 3 years of age, respectively (Table 1).

### 3.2. Nutritional Quality of Plant Proteins for Infants

For the assessment of the quality of proteins in a food or a diet, the accepted method is the chemical scoring method (PD-CAAS and DIAAS) [28]. The IAA score is the ratio of the content of each IAA (mg/g protein) in the food ingredient or formulation to the reference pattern of IAA (mg/g protein) [30,31]. In addition, the scoring approach considers the content of bio-available IAA in food and diet that represents the dietary intake, which is made available to the organism for metabolism after digestion and absorption and is oriented to sequential anabolic and catabolic pathways. Bioavailability is considered to be associated with digestibility that measures digestive losses expressed as a proportion of ingested nitrogen or amino acids that is absorbed in the intestine following protein consumption.

The PDCAAS and DIAAS values are computed by correcting the lowest chemical score of one of IAA by either the protein digestibility or the specific digestibility of the limiting IAA [30,31].
Chemical amino acid score = ((IAA in mg/g test protein)/(IAA in mg/g reference pattern)).(1)
PDCAAS = Limiting IAA Score (AAS) × weighted protein digestibility for the food.(2)
DIAAS = Limiting IAA Score (AAS) × weighted IAA digestibility for the food.(3)

The lowest PDCAAS or DIAAS value is applied to the protein. A protein score below 1 indicates that at least one IAA is limiting in the food or diet and a score of 1, that there is no limiting IAA in the food or diet. Scores are truncated to 100 percent.

For the different plant protein sources, their total IAA content is more often lower than in animal protein, accounting for about 30–40% of amino acids in plants and 40–50% in animal protein [27,33,34]. Moreover, the digestibility of plant proteins (70–90%) is also generally lower than that of animal proteins (90–99%), and the difference is even greater when vegetable proteins are consumed as complex flours or whole seeds [35,36,37]. From the available data on IAA content and digestibility of protein from different plant sources, calculation of PDCAAS using the IAA reference pattern for either adults, 1–3-year-old children, or the composition of human milk, shows that the PDCAAS values are all below 1, except for the upper value of the range for soy for adults (Table 2). Moreover, due to the higher metabolic demand for IAA, the PDCAAS values are lower for infants compared to adults and are also lower when using the composition of human milk, compared to the IAA reference pattern for 1–3-year-old infants.

As plant proteins are less rich in IAA than animal proteins and have a significantly lower digestibility, their nutritional quality is generally lower, particularly in infants and children due to higher quality demand compared to adults. It is therefore necessary to fortify the foods with the limiting IAA that are generally for pulses, the sulphur IAA (SAA), and sometimes the branched-chain IAA (BCAA), and for cereals lysine and threonine. Tryptophan may also be limiting for some plant sources.

### 3.3. Manufacturing Hydrolyzed Rice Proteins

Until recently, rice flour has mostly been processed by the industry to collect starch, the residual proteins being discarded or used for animal feeding [38]. Nowadays, rice proteins are considered a valuable raw material by manufacturers, possessing a high nutritional profile and being hypoallergenic. The rice protein content may vary with genotype, climate, and cultural practice, but usually represents 7–10% of the grain weight [39].

Rice has four types of proteins classified according to their solubility: albumin (water-soluble), globulin (salt-soluble), glutelin (alkaline-soluble), and prolamin (alcohol-soluble) [40]. They are mainly stored in two different cellular organites called protein bodies (PB): PB-type I (PB-I) and PB-type II (PB-II) [41].

Rice proteins come from either rice bran or broken rice kernels. Rice bran is a floury material made of seed, pericarp, coat, aleurone, and some embryo, hull, and endosperm fragments (Figure 1). Broken rice kernels are rich in starch and contain endosperm storage proteins only [42].

The protein content is higher in rice bran than in broken rice kernels (respectively 12% and 7% of weight) and rice bran proteins have a higher nutritional quality. In Han’s study (2015), protein efficiency ratio (PER), net protein retention (NPR) and net protein utilization (NPU) were much higher in rice bran (PER: 2.39, NPR: 3.77 and NPU: 70.7) than in rice endosperm (PER: 1.96, NPR: 3.26 and NPU: 61.4) [43]. However, rice bran proteins are strongly bonded and form complexes extremely insoluble in water, so that the extraction process of rice bran proteins is a real industrial challenge [44]. Besides, rice bran contains high level of lipids, providing a substrate easy to hydrolyze by lipase-enzymes, which may induce a rancid flavor in the rice [42]. Therefore, rice bran enzyme activity needs to be stabilized by heat, pH, or defatted treatment, all processes that may denature proteins which aggregate and become harder to purify [45,46].

Broken rice kernels are fractions of the rice endosperm, which mainly contain glutelin (alkaline soluble) and starch. Therefore, extracting proteins from the endosperm part is easily achievable under alkaline conditions (Figure 2).

#### 3.3.1. Extraction

Rice proteins are highly water insoluble due to intermolecular disulfide linkages and high molecular weights [44,47], complicating extraction and purification. Rice endosperm proteins mainly consist of glutelin, from which extraction is carried out efficiently (97%) in alkaline conditions, with 0.1 N sodium hydroxide or potassium hydroxide [41,48]. A thorough control of alkaline solvent parameters is needed to avoid extracting non-protein components that may co-precipitate with proteins, lowering the protein purity of the resultant ingredient, increasing Maillard reactions (due to high pH) with formation of dark colored (melanoidin-based) products and of toxic compounds such as lysinoalanine [49,50].

The main component of the rice endosperm is starch, which can be solubilized and removed with enzymes such α-amylase, glucoamylase and pullulanase, to separate proteins in rice flour [44], a process leading to protein isolates, containing more than 90% of proteins. However, compared to the starch hydrolyzation process, the alkaline one leads to proteins with higher digestibility and bioavailability, as a result of structural changes in the non-digestible PB-I triggered by alkaline conditions [51,52].

#### 3.3.2. Hydrolyzation

Alkaline extraction increases the solubility of rice proteins and their enzymatic hydrolyzation enhances their functionality in the formulation of processed food. Enzymes such as proteases increase the number of ionizable groups with a lower molecular weight, thus the solubility of the raw material. A study comparing rice endosperm protein concentrate and hydrolyzed rice endosperm protein, clearly demonstrates that hydrolyzed rice endosperm protein was almost completely soluble in water at pH 2 to 8 while their intact counterpart rice endosperm protein concentrate displayed very low solubility values [53].

In infant formulas, the rice protein raw material thus requires two steps, first to separate proteins from starch and second to obtain a perfectly water-soluble product.

## 4. The Development of Hydrolyzed Rice Protein Formulas (HRPF) during CMPA

This analyzes how HRPFs could be legally developed in accordance with the regulation of the European Union, the manufacturing of different brands, and the evidence available on the adaptation of this type of formula for infant feeding.

### 4.1. Legal Framework

In the European Union, the only sources of protein allowed in infant and follow-on formulas are cow’s milk proteins (CMP), goat’s milk proteins (since 2013), soy protein isolates and hydrolyzed proteins [54].

Under EU law, the term “foods for special medical purposes” (FSMP) refers to a category of foods intended for a particular diet, which are specially treated or formulated to meet patient needs to be used only under medical supervision. They are intended to constitute the exclusive or partial diet of either patients with a low ability to absorb, digest, assimilate, metabolize, or excrete ordinary foods, ingredients, and metabolites, or patients whose health status determines other specific nutritional needs that cannot be met by a change in the normal diet or by a diet consisting of food or a combination of the two.

These indications are generated by generally accepted scientific evidence and their efficacy and safety in young children must be demonstrated by high-quality scientific studies [5].

Dietary foods containing hydrolyzed rice protein (HRP) are therefore in the FSMP category, where they have been used since the early 2000s in several European countries (Italy, Spain, and France) for the treatment of CMPA.

All the hydrolyzed rice protein formula (HRPFs) ingredients are of plant origin, except for vitamin D3 (cholecalciferol). HRPFs do not contain phytoestrogens and the rice is not genetically modified. The content of arsenic, heavy metals, and pesticides is strictly regulated for food in children under 3 years of age, in compliancy with the EU Directive 2013/46 of 28 August 2013 amending directive 2006/141 [54]; manufacturers are required to comply within safe limits as set out and described by The ESPGHAN Nutrition Committee recommendations in 2015 [55]. Since 2016, the maximum rice inorganic arsenic content for food in children under the age of 3 is 0.10 mg/kg (two times lower than that applied to white rice) [56].

### 4.2. HRPF Market Availability

Literature data indicate that 5 different HRPF brands have been tested in Italy (1), the United States (1), Spain (1), and France (2) [3].

In Italy, Plasmon Risolac^®^ (Kraft Heinz Co, Chicage, IL USA), launched in 2000, is currently available in a unique formula (Risolac^®^) for infants and young children from 0 to 3 years old. The Italian health authorities define this formula as an FSMP for children allergic to CMP or soy. In Spain, Blemil Plus Arroz Hidrolizado^®^ 1 and 2 (Ordesa, Barcelona, Spain) were launched in 2008. Contrary to Risolac, these formulas are reimbursed by the Spanish health system in the category “lactose-free hydrolysates’’. They are prescribed in cases of CMPA, primary or secondary lactose intolerance, chronic or acute diarrhea, and refeeding after an episode of acute diarrhea. In France, Modilac Expert Riz^®^ 1 and 2 (Sodilac, Paris, France) were launched in 2009. Their composition is identical to that of Blemil Plus Arroz^®^ 1 and 2. Modilac Expert Riz AR^®^ (Sodilac, Paris, France) was afterwards launched in 2013 CMPA with regurgitations. Modilac Expert Rice^®^ 3 (Sodilac, Paris, France) was launched in 2016 for young children aged from 1 to 3 years. Novalac Riz^®^ (Novalac, Paris, France) was launched in 2012 for infants and young children, from 0 to 3 years. The HRPF as here mentioned above are not reimbursed by French health insurance.

Picot Riz^®^ 1 and 2 (Lactalis nutrition health, Laval) launched in 2012, Premiriz^®^ 1, 2, and 3 (Candia-Baby, Paris, France), and Baby Mandorle rice^®^ 1 and 2 (La Mandorle, Paris, France) are not proposed by manufacturers as a treatment for CMPA and will not be discussed further in this document due to a total absence of clinical trials. An additional HRPF (Ross Products, Abbott) evaluated in a 2006 clinical trial [57] has not been commercialized to date.

It should be noted that a formula indicated for the treatment of CMPA should be tolerated by at least 90% of children allergic to CMP, with a confidence level of 95% [58,59].

### 4.3. HRPF’s Occupancy in the CMPA Market

CMP-eHF accounts for 48.4% of the total FSMP used for CMPA in France, HRPF 39.3%, and free amino acid preparations for 12.3%. These figures represent respectively 6.0%, 4.9%, and 1.5% of all formulas used for children aged 0 to 3 [60].

HRPFs are currently marketed mainly in Italy, France, and Spain. They are found in a growing number of regions such as North Africa, the Middle East, and South America, but are still not available in many countries in Europe, the United States, Canada, Australia, and New Zealand. Existing guidelines recommend the use of a CMP-eHF (from whey or casein) as the first care of children with CMPA, while HRPFs are not mentioned or only as a second line, as they do not or are not available in many countries. HRPFs are growing in popularity because they have proven to be effective and safe, have good acceptability, and are cheaper than CMP-eHF [61,62]. The cost of an HRPF is close to that of standard premium infant formulas or a follow-on. In comparison, a CMP-eHF, which is nutritionally adequate and well tolerated by children allergic to CMP and other foods, can have some drawbacks such as a bitter taste [63], a higher cost (2 to 3 times that of standard premium formulas), and contain a potential risk of anaphylaxis in some children. AAFs, offered in severe clinical situations or in children who do not respond to the CMP-eHF, are safe but more expensive, 2 to 3 times the cost of CMP-eHF [62,64]. The ESPGHAN Committee on Gastroenterology stated in 2012 that the use of an HRPF is an option if it has proven its efficacy and safety in infants with CMPA [65].

Based on studies published to date, HRPF are effective in children with CMPA and provide nutritional security. No data is available on the use of HRPF in case of an allergy to CMP-eHFs, which currently requires the use of AAFs, as well as on their efficiency on bone mineralization. The effect of HRPF on the duration of the CMPA, compared to the effect of CMP-eHF, remains insufficiently documented.

## 5. Nutritional Properties of HRPFs

### 5.1. Energy, Protein, Lipid, and Carbohydrate Content of HRPF

The energy content and lipid composition of HRPFs is comparable to that of standard infant or follow-on formulas [3] (Table 3).

The nutritional value of a protein depends on its digestibility coefficient:
Digestibility coefficient = nitrogen ingested − excreted nitrogen/nitrogen ingested × 100(4)

The digestibility coefficient of rice protein is lower than that of CMP: 93 vs. 100% [69]. Because of this variation, the protein content of HRPFs is slightly higher than the current average protein content of standard formulas [70] (Table 3). The nutritional value of a protein also depends on its amino acid composition and although rich in essential amino acids, rice grain contains limited amounts of lysine, threonine, and tryptophan [71]. However, addition of these three amino acids brings the HRPF aminogram close to that of human milk protein which meets the amino acid needs of infants [72] (Table 3).

As mentioned previously, rice proteins undergo enzymatic hydrolysis, a process that allows the major rice proteins (80% glutelin and 10% globulin) to be water-soluble [57,73]. In HRPFs, peptides generally possess a low molecular weight (MW) (Table 3).

Currently, HRPFs are lactose-free; a lactose-containing formula used in a study [74] is no longer available. The carbohydrate fraction is mainly composed of dextrin-maltose, corn starch, and different kind of syrups (Table 3).

### 5.2. Nutritional Efficiency of HRPFs

In two different studies, healthy infants who were fed with HPRF from birth until complementary feeding showed appropriate growth, demonstrating the normal nutritional efficiency of these formulas (Table 4).

### 5.3. Acceptability and Palatability of HRPFs

Overall acceptance of HRPF was good in the Lasekan et al. (Ross) [57], D’Auria et al. (Risolac^®^) [76], Fiocchi et al. (Risolac^®^) [77], and Girardet et al. (Modilac^®^) studies [74]. The Study of Reche et al. (Blemil Arroz^®^) [67], showed an identical infant rejection in HRPF and CMP-eHF formulas (i.e., 2/46 infants against HRPF and 2/46 infants against CMP-eHF). The study by Vandenplas (Novalac^®^) [68] showed that 18.8% of parents felt that their infant did not like or accept the study formula, leading to a drop-out in 3 of 40 included infants: a possible explanation for this could be the high level of rice protein hydrolysis, increasing the taste of bitterness [78].

A double-blind study of Pedrosa et al. [79] tested the appetite (taste, smell, texture) of 12 different formulas in 50 randomized adult subjects: these tests demonstrated the clear superiority of HRPF and soy formula over different CMP-eHF. In children, the palatability of HRPF was found to be superior to that of CMP-eHF [80].

### 5.4. Digestive Tolerance of HRPFs

Protein hydrolysis increases the formula osmolarity, which may enhance regurgitation due to the delay in gastric emptying. This could also induce softer and more frequent stools, which may be of a greenish color due to more intraluminal water secretion [78]. To limit these effects, HRPF is thickened with corn starch in Risolac^®^, Blemil^®^, and Modilac^®^ formulas, and with pectin in the Novalac^®^ formula. The Lasekan et al. study [57], in healthy infants, showed that the digestive tolerance of HRPF was good and comparable with non-hydrolyzed standard cow’s milk protein formula (SF) tolerance. Furthermore, the HRPF did not modify the quantity of stools or consistency. Infants fed HRPF tended to have less regurgitation and vomiting than infants fed SF. In the D’Auria et al. study [76] in infants with CMPA, HRPF did not induce any adverse effects. The Vandenplas Study [68] showed that only 5.3% of infants with CMPA had normal stools upon study inclusion, while normalization of stools was observed in 52.6% of infants after one month of HRPF feeding and 77.8% after three months. However, this was an observational study; it is not possible to formally validate these findings.

### 5.5. Allergenic Tolerance of Rice and of HRPFs

Rice grain has low allergenicity, and rice protein allergy is uncommon in publications from Western countries [3,81]. Rice triggers adverse reactions in less than 1% of allergic children and it is considered as the least allergenic cereal [82]. Nonetheless, rice proteins may be the cause of food protein-induced enterocolitis syndrome (FPIES), a non-IgE-mediated condition, but less frequently than CMP or soy protein. The time-lapse to diagnosis is longer and the severity of symptoms is higher than those for CMPA [55,83]. Children with rice-induced FPIES are more likely to develop FPIES to other foods (oats, barley, wheat, and other non-grain foods) than those with cow’s milk or soy FPIES. From 1963 to 2009, 42 cases of rice FPIES were reported [84]. In all cases, the whole rice grain had been consumed before the reaction and an HRPF was never reported as a FPIES trigger. To date, there are no published cases of reported allergy to am HRPF, even though personal communication and experiences indicate a limited number of cases.

The allergenicity of an HRPF (Risolac^®^) was tested by Piacentini et al. [85] in 130 young guinea pigs fed ad libitum for 37 days either with the HRPF or with a standard CMP formula (SF). After the sensitization period, guinea pigs were injected intravenously with isolated whole proteins (CMP and rice proteins) or ultra-centrifugal formulas. Specific IgGs against beta lactoglobulin, caseins, and whole rice proteins were measured. In the CMP-based SF-fed group, the injection of beta-lactoglobulin, casein, or whole ultra-centrifugated CMP resulted in significantly more reactions than those in the HRPF-fed group that received injection of the same proteins (p-0.001). In the HRPF-fed group, no reaction was observed after the test with ultracentrifugated HRPF, and only two benign reactions occurred after the test with rice protein. Extremely low levels of specific IgG antibodies against rice proteins were noted in all groups, including animals fed HRPF, with no significant difference between the groups.

## 6. HRPFs in the Treatment of CMPA

HRPFs are used mostly during CMPA, which expectedly represents its main field of investigation. Clinical trials focused mainly on hypoallergenicity and on nutritional reliability, were measured by their impact on growth.

### 6.1. Allergenic Efficiency of HRPFs in CMPA

Most studies in children with CMPA focused on IgE-mediated CMPA, with a diagnosis based on serum determination of specific IgEs by radioallergosorbent tests (RAST), skin prick tests (SPT), and oral food challenge (OFC). Non-IgE-mediated CMPA can only be diagnosed by OFC. Two studies [68,86] included both IgE-mediated CMPA and non-IgE-mediated CMPA. In these studies, the HRP formulas were well tolerated by at least 90% of children allergic to CMP (Table 5).

### 6.2. Nutritional Efficiency of HRPF in CMPA

#### 6.2.1. Growth evolution and protein nutritional status

Growth in the first months of life to two years of age in infants with CMPA fed a HRPF (Table 6) was evaluated in five studies [3]. A significant growth retardation existed in most studies [68,81,87].

In the study by Savino et al. [66], the HRPF contained the same energy content (67.8, 67.6 and 67.6 kcal/100 mL, respectively) but 15 to 20% less protein (1.5 g/100 mL) than the soy formula (1.8 g/100 mL) and CMP-eHF (1.9 g/100 mL), a difference that may explain the slightest weight gain in children fed with HRPF [73]. However, during the exclusive bottle-feeding period, weight gain was comparable in the HRPF and the control groups. A role of the milk and dairy elimination diet on growth parameters was suggested by the lower growth rate observed in children with CMPA after the age of nine months, compared to children without CMPA in the control group [7,8].

Overall, all seven studies show with an HRPF an appropriate growth pattern in healthy children and a catch-up growth in those with CMPA. Still, the studies focused on small samples and were of relatively short durations.

#### 6.2.2. Bone mineralization

Two studies [57,86] found similar plasma concentrations of calcium, magnesium, and alkaline phosphatase in healthy children receiving HRPF or an SF. These are only indirect markers of bone health and no study has been carried out so far on the impact of HRPF feeding during CMP on fracture incidence, bone density, or on the bioavailability of minerals and micronutrients present in HRPFs.

### 6.3. Efficiency of HRPF in the Acquisition of Tolerance

The duration of CMPA according to the formula used was discussed in three studies, with methodologic issues preventing any conclusion in that matter (Table 7).

## 7. Conclusions

HRPFs now represent, in some countries, a valuable option for feeding infants with CMPA. The legal frame of FSMPs regarding both CMPA formulas and the quality of rice proteins allows feeding infants and children without the risk of nutritional deficiency. The low availability of these formulas explains a poor presence in guidelines referring to feeding during CMPA. Based on the current evidence, they seem appropriate to restore a normal growth and may be used as a first choice during CMPA. Evidence is lacking on their long-term use in infants and children, especially for bone mineralization, on their role in the acquisition of tolerance in allergic children, and on their exact place in complex conditions associated with CMPA, such as allergy to hydrolysates and multiple food allergy.

## Figures and Tables

**Figure 1 nutrients-12-02654-f001:**
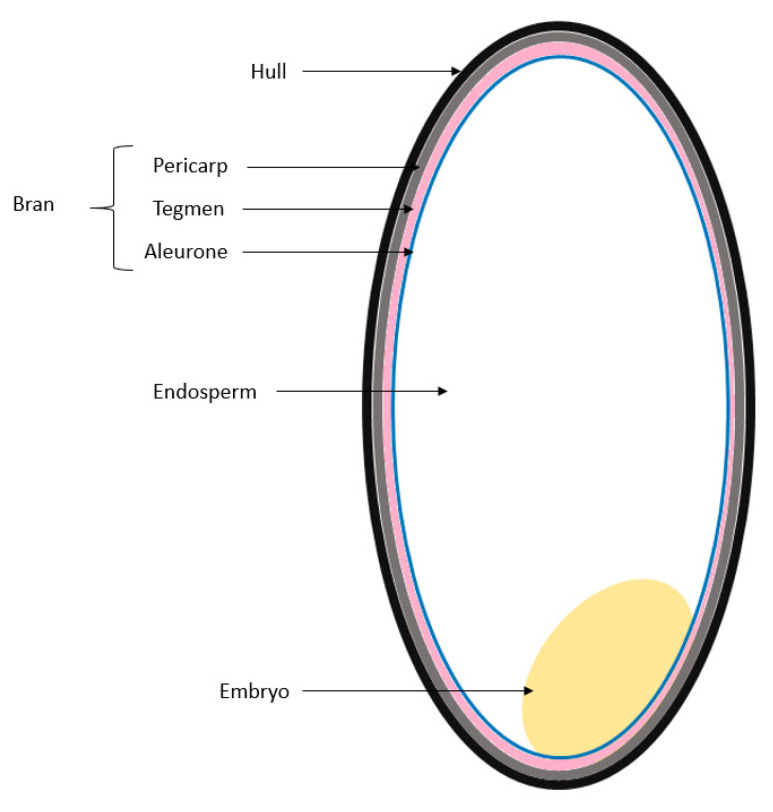
Structure of the rice grain.

**Figure 2 nutrients-12-02654-f002:**
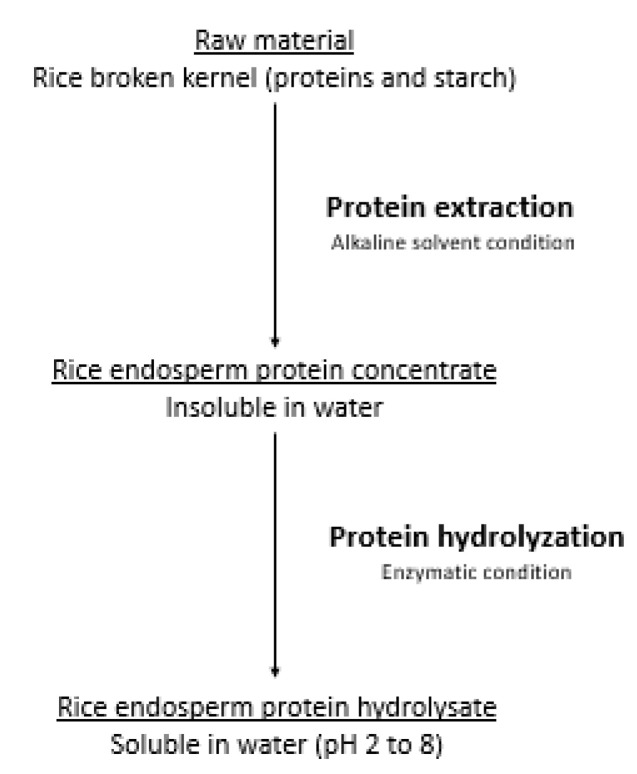
The two steps manufacturing process of the hydrolyzed rice protein raw material.

**Table 1 nutrients-12-02654-t001:** Indispensable amino acids (IAA) reference pattern for infants and children derived from mother’s milk composition and calculated by the factorial method for young children 1–2.9 years in comparison to adults [27,32].

	Average Protein Requirement	IAA Reference Pattern (mg/g Protein)								
	g/kg/day	His	Ile	Leu	Lys	SAA	AAA	Thr	Trp	Val
Human milk	-	21	55	96	69	33	94	44	17	55
1–2.9 years	0.86	18	31	63	52	26	46	27	7.4	42
Adult	0.66	15	30	59	45	22	38	23	6	39

SAA: Sulphur-Containing amino acid. AAA: Aromatic amino acid.

**Table 2 nutrients-12-02654-t002:** Limiting IAA, protein digestibility and PDCAAS values of different plant protein sources, for adult and young children, or calculated from human milk composition.

Protein Source	First Limiting IAA	Protein Digestibility %	PDCAAS		
			Adult	1–3 Years	Human Milk
Soja	SAA	75–90	0.8–1.0	0.7–0.8	0.6–0.7
Pea	SAA	75–90	0.8–0.9	0.7–0.8	0.6–0.7
Chickpea	SAA	75–85	0.8–0.9	0.6–0.7	0.5–0.6
Lupine	valine	85–90	0.8–0.9	0.7–0.8	0.6–0.7
Lentil	SAA	75–80	0.6–0.7	0.5–0.6	0.4–0.5
Peanut	Lysine	85–90	0.6–0.7	0.5–0.6	0.4–0.5
Wheat	Lysine	75–85	0.4–0.6	0.3–0.5	0.2–0.3
Rice	Lysine	75–85	0.5–0.7	0.4–0.6	0.3–0.5

**Table 3 nutrients-12-02654-t003:** Energy, protein, lipid, and carbohydrate content of HRPFs.

Formula	Energy/100 mL	Proteins/100 mL	Peptides Molecular Weight (MW)	Addition of Free Amino-Acids			Lipids	Carbohydrates/100 mL	
				Lysin	Threonine	Tryptophan			
Risolac^®^ 0–3 years [66]	69 kcal	2.1 g	44%: MW < 1000 Da 43%: 1000 Da < MW < 2000 Da 13%: 2000 Da < MW < 4000 Da	Yes	Yes	Yes	Similar to standard formulas	Lactose free	Dextrin-maltose: 5.3 g Corn Starch: 0.5 g Glucose + saccharose syrup: 1.5 g
Blemil Arroz 1^®^ Modilac Expert Riz 1^®^ [67]	71 kcal	1.7 g	10%: Free amino acids 26.8%: MW < 300 Da 29.9%: 300 Da < MW < 1000 Da 35.2%: 1000 Da < MW < 5000 Da	Yes	Yes	Yes			Dextrin-maltose: 6 g Corn Starch: 1.6 g
Blemil Arroz 2^®^ Modilac Expert Riz 2^®^ [67]	69 kcal	2 g		Yes	No	Yes			Dextrin-maltose: 6.4 g Corn Starch: 1.7 g
Novalac^®^ 0–3 years [68]	68 kcal	1.8 g	95%: MW < 1000 Da	Yes	No	Yes			Dextrin-maltose: 5.7 g Corn Starch: 1.9g
Ross Formula [57]	68 kcal	1.9 g	Unknown	Yes	Yes	No			40% rice syrup + 60% saccharose syrup: 6.7 g

**Table 4 nutrients-12-02654-t004:** Healthy infants fed with HPRF had a normal growth.

Study	Patients	Type of Study	Intervention	Outcomes
Lasekan et al., 2006 [57]	65 healthy infants (without CMPA) Age: 0 to 16 weeks	Randomized double-blind trial	HRPF (Ross formula) or standard formula for 4 months	Height, weight, BMI, and cranial girth within normal limits No difference between groups
Girardet et al., 2010 [74]	78 healthy full-term infants Age: <1 month	Open multicenter prospective study	Lactose-containing HRPF (Modilac^®^) from the 1st month to the age of 4 to 6 months (after starting complementary feeding)	Average daily weight gain: 23.2 ± 4.3 g (PP population), with no difference with WHO standards [75] Height, weight, and BMI z-scores (intent to treat population): between 1.1 and −0.5 DS during the study period

**Table 5 nutrients-12-02654-t005:** Allergenic efficiency of HRP formulas in CMPA infants.

Study	Patients	Type of Study	Study Duration	Intervention	Outcomes
Fiocchi et al., 2003 [77]	18 infants CMPA confirmed by a double-blind placebo-controlled food challenge (DBPCFC) Age: 1–9 years (average 5 years)	Clinical trial	1 test	HRPF (Risolac^®^)	Skin Prick Test: CMP: positive in all children Soy: positive in all children Rice: positive for 8/18 children HRPF: positive for 2/18 children Specific IgEs: CMP: positive in all children Soy: positive in 13/18 children Rice: positive in 7/18 children HRPF: permanently negative Double-blind placebo-controlled food challenge (DBPCFC) with HRPF: negative in all cases
Fiocchi et al., 2006 [69]	100 infants CMPA confirmed by DBPCFC Age: 3.2 ± 2.93 years	Prospective study	1 test	HRPF (Risolac^®^)	Skin Prick Test: Cow’s milk and/or CMP fraction: positive in 87/99 children Rice: positive in 4/90 children HRPF: positive in 4/86 children Specific IgEs > 0.35 KU/L: Cow’s milk and/or for a CMP fraction: in 92/95 children Rice: in 21/91 children HRP: in 4/91 children Rice specific IgEs: Rice: positive in 21/91 children (Pharmacia—Upjohn Diagnostic) and in 70/96 children (immunotransfer) HRPF: weakly positive in 6 children DBPCFC with HRPF was always negative.
Reche et al., 2010 [67]	92 infants CMPA IgE-mediated confirmed by a positive Oral Food Challenge (OFC) Age: average 4.3 months (1.1 to 10)	Prospective, open and randomized clinical trial	2 years	46 fed a HRPF (Blemil Arroz^®^/Modilac Expert Riz 1^®^) 46 fed a CMP-eHF	HRPF: well tolerated in all children CMP-eHF: 1 child developed allergy to this -CMP-eHF Evolution of number of children remaining allergic: similar in both groups.
Vandenplas et al., 2014 [68]	40 infants CMPA confirmed by OFC CMPA IgE-mediated or not Age: average 3.4 month (1 to 6)	Prospective trial	6 months	HRPF (Novalac Riz^®^)	Significant decrease of the allergy symptoms after 1 month Benefit confirmed after 3 and 6 months Clinical tolerance was assessed with the symptom-based score (SBS) [81], now published as the COMISS score [82,83]

**Table 6 nutrients-12-02654-t006:** Growth and weight evolution and protein nutritional status in children affected with CPMA or healthy children (one study).

Study	Patients	Type of Study	Height and Weight z-Scores at Inclusion	Intervention	Outcomes
D’Auria et al. 2003 [76]	16 infants CMPA + atopic dermatitis (DBPCFC or an open OFC or an open test) Age: 6–14 months	Observation	Weight: HRPF: −0.30 (−0.34) Soy formula: −0.21 (−0.14) Height: HRPF: −0.10 (−0.21) Soy formula: −0.12 (−0.23)	HRPF or soy formula	Weight: HRPF: 0.09 (−0.08). Soy formula: 0.11 (0.15). Height: HRPF: 0.07 (0.12) Soy formula: 0.27 (0.37). Protein nutritional status: Blood markers of protein homeostasis (albumin, pre-albumin, total plasma proteins, urea): similar in both groups
Savino et al., 2005 [66]	58 infants CMPA + atopic dermatitis 30 controls (without CMPA) Age: 1–24 months	Prospective, non-randomized, mono-centric, open		HRPF (Risolac^®^) Or a soy formula Or CMP-eHF Or free diet in control group	Weight; Weight z-score were similar in the 3 groups with CMPA during the first 2 years. Weight gain was smaller in the HRPF group (*p* 0.025) vs control group.
Agostoni et al., 2007 [87]	93 infants CMPA 32 controls Age: 6–12 months	Multi-center-forward, randomized, comparative, open	Weight: RHPF: −0.30 (−0.34) Control: −0.10 Height: RHPF: −0.21 Control: −0.12	soy formula (n-32), CMP-eHF (n-31) and HRPF (Risolac^®^) (n-30)	Weight: RHPF: −0.09 Control: 0.07 Height: RHPF: 0.11 Control: 0.27
Reche et al. 2010 [67]	92 infants CMPA (skin prick tests) Age: 1.5–9 months	Prospective open, randomized		CMP-eHF and HRPF	Weight: similar Height: similar
Vandenplas et al., 2014) [68]	42 infants Age: 3.4 ± 1.5 months		Weight: HRPF: −0.7 ± 1.0 Height: HRPF: −0.1 ± 1.0	HRPF	Weight: HRPF: −0.1 ± 0.9 Height: HRPF: −0.1 ± 1.1
Lasekan et al., 2006 [57]	65 infants Healthy infants (without CMPA) Age: 0 to 16 weeks	Randomized double-blind trial		HRPF (Ross formula) or standard formula for 4 months	Weight: similar Height: similar Protein nutritional status: Plasma protein concentrations, particularly for total plasma proteins, serum albumin, and pre-albumin/transthyretin: comparable in both groups

**Table 7 nutrients-12-02654-t007:** The duration of CMPA in infants fed with HRPF.

Study	Patients	Type of Study	Intervention	Outcomes
Reche et al., 2010 [67]	92 infants CMPA IgE-mediated confirmed by a positive OFC Age: average 4.3 months (1.1 to 10)	Prospective, open and randomized clinical trial	46 fed a HRPF (Blemil Arroz^®^/Modilac Expert Riz 1^®^) 46 a CMP-eHF for 2 years	Percentage of children becoming tolerant: similar with the HRPF and the—CMP-eHF after 12, 18, and 24 months of feeding
Terraciano et al., 2010 [88]	72 infants CMPA Age: 14.1 ± 8.6 months	Prospective cohort	Fed with CMP-eHF or soya formula or HRPF For 26 months (median duration)	Time before tolerance was acquired (median duration of the disease): CMP-eHF group: 56 months (IC 95% not applicable) (average—ES: 40.2–4.8 months) Soy formula group: 28 months (IC 95% 11–37) (average—ES: 24.3–2.6) HRPF group: 20 months (IC 95% 10–33) (average—ES: 24.3–3.6) This beneficial effect was not observed in polysensitized children
Berni Canani et al., 2013 [86]	260 infants CMPA confirmed by DBPCFC with milk Age: 1 to 12 months	Multicenter retrospective observational study	71 fed with a CMP-eHF- Lactobacillus Rhamnosus GG (LGG) 55 with a CMP-eHF, 46 with a HRPF (Risolac^®^) 55 with a soy formula 33 with an amino-acids-based formula (AAF)	Percentage of patients having outgrown CMPA after 12 months: Similar with the CMP-eHF, the HRPF, soy formula and the AAF Significantly shorter in-CMP-eHF-LGG group (OR 4.8; 95% CI 2.2–10.5; *p* 0. 001)

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
