# Peer review of "Hydrolyzed Rice Protein-Based Formulas, a Vegetal Alternative in Cow’s Milk Allergy"

_nutrients, 2020, doi:10.3390/nu12092654_

Round 1
Reviewer 1 Report
This manuscript addresses a very interesting and current topic in the field of infant nutrition. Nevertheless, as the manuscript currently stands, I am not sure at all that it crosses the threshold for the level of contribution required for Nutrients. It is well-written for a book chapter, but not for a peer-reviewed journal article type. Thus, for me, improvements in the manuscript should be made in terms of increasing its contribution to the literature. Overall, I found many limitations in the manuscript, which I explain below (trying to provide solutions where possible).
Mayor:
- Sections 4 & 5. The development of hydrolyzed rice protein formulas (HRPF) & HRPFs in the treatment of CMPA. One of my main concerns with this paper is that most of the text is similar or even identical to what you already have published previously (Bocquet, A., Dupont, C., Chouraqui, J. P., Darmaun, D., Feillet, F., Frelut, M. L., ... & Simeoni, U. (2019). Efficacy and safety of hydrolyzed rice-protein formulas for the treatment of cow's milk protein allergy. Archives de Pédiatrie, 26(4), 238-246.). These sections should be fully rewritten. Thus, I strongly recommend to simplify the existing text and include new text that clearly adds to your previous published research.
- The manuscript is written as a book chapter. Most of the paper is describing in short sentences the results of other authors. No link between sentences to create a story line. Also, it is not clear what research questions you want to address with your manuscript. In particular, at the end of the introduction section I strongly suggest adding an explanation of why you are doing the review and ideally a list of questions you want to answer with the results of your review. For example. We aimed to….., we wanted to address the following questions in our review: 1), 2, 3….or alternatively, you can quickly introduce how the article is structured: for example….this article (1) provides an overview of the nutritional hazards related to CMPA, (2) discuss the use of plant proteins in infant nutrition and how rice based infant formulas are manufactured, (3) describes the legal framework, market characteristics and nutritional properties of HRPF, And (4) describes the use of HRPF for CMPA treatment. Either way, no need to strengthen why your study is important and how you intend to contribute to the literature.
- Difficult to read and to extract conclusions. As the paper is very complete and is touching different topics, I strongly suggest to add a short introduction in each section and subsection to show to the reader what is going to read. Moreover, at the end of each section a summary of main conclusions is recommendable. .
Minor
Article type
- In my opinion this is a review paper not an article type as there is no orginal research.
Format
- No numbers or bullets and different font sizes in in the subsections and subsubsections. Please see some examples as follows:
- Line 37: 1. Height and weight gain should be 2.1
- Line 184: 3. Manufacturing hydrolyzed rice proteins should be 3.3
- Line 212: Extraction should be 3.3.1 and smaller font size
Introduction
- Introduction is very weak and too short. As a potential solution, I suggest authors to move and rewrite part of the subsections to the introduction (e.g., 4.1 Legal framework and 4.2 market availability of HRPS).
Section 2. Cow’s milk protein allergy (CMPA) and its nutritional hazards
- Line 36- Pls add a short introduction about the four subsections you are going to discuss ….. Height and weight gain, Micronutrients and macronutrients deficiency, Feeding difficulties, The need for a precise dietary follow-up…..
- Line 37. Suggest changing Height and weight gain by Height and weight gain problems
- Line 55. Suggest changing deficiency by deficiencies
- Lines 63-86: It is difficult to follow the read. I suggest to better structure this part and not only describe in each sentence the research from other authors.
Section 3. Plant proteins in infant nutrition
- I founded this part very innovative and I believe that clearly contributes to existing literature, especially subsections 2 and 3.
- Please add an introduction describing what you are going to deal with in this subsection (e.g., …….Protein and amino acid requirements for infants, the importance of the nutritional quality of plant proteins and how rice proteins are hydrolized to fulfill the main challenges of plant proteins….)
- At the end of the chapter I would suggest to summarize the main conclusions of the subsection.
Overall, I found that you are dealing with an interesting topic, but as explained earlier your manuscript does suffer from many limitations. I hope my comments can help you in making this a better, stronger manuscript. Good luck!
Author Response
Dear Mrs., Mr.
First, I would like to thank you for correcting our paper and for your accurate advices that they’ve been considered very seriously.
Mayor :
- Sections 4 & 5. The development of hydrolyzed rice protein formulas (HRPF) & HRPFs in the treatment of CMPA.
All the text with >2% similarity with the previously published paper has been rewritten and some new information has been added (Walter C et al., Milk and Health, N Engl J Med 2020;382:644-54)
- The manuscript is written as a book chapter.
The introduction has been rewritten to explain more accurately the context of the paper, the expanding disease, CMA, the potential challenges health care professionals have to deal with, and the development of plant proteins, and more specifically, rice protein as new resources for industry. We did that in Abstract and Introduction, and added a few words of introduction in each chapter .
- Difficult to read and to extract conclusions.
Introduction and conclusion have been added to each section.
Minor
Article type
In my opinion this is a review paper not an article type as there is no original research.
We agree with you
Introduction is very weak and too short. As a potential solution, I suggest authors to move and rewrite part of the subsections to the introduction (e.g., 4.1 Legal framework and 4.2 market availability of HRPS).
The introduction has been rewritten to give a more precise context.
Section 2. Cow’s milk protein allergy (CMPA) and its nutritional hazards
Line 36- Pls add a short introduction about the four subsections you are going to discuss ….. Height and weight gain, Micronutrients and macronutrients deficiency, Feeding difficulties, The need for a precise dietary follow-up…..
Introduction and conclusion have been added to each section.
Line 37. Suggest changing Height and weight gain by Height and weight gain problems
Done.
Line 55. Suggest changing deficiency by deficiencies
Done.
Lines 63-86: It is difficult to follow the read. I suggest to better structure this part and not only describe in each sentence the research from other authors.
This section has been entirely rewritten
Section 3. Plant proteins in infant nutrition
I found this part very innovative and I believe that clearly contributes to existing literature, especially subsections 2 and 3.
Thank you for this comment.
Kind regards,
Reviewer 2 Report
This is a very comprehensive and well written review of hydrolyzed rice protein based formula for cow's milk allergy in infants and children. My comments consist of minor edits suggested below.
Line 10. The nutritional quality of rice proteins allows their use, either exclusive or supplemented by certain 10 amino acids. (insert full stop here)
Line 11. are adapted to the diet of allergic children to…
Suggest: are adapted to the diet of children allergic to…
Line 14. and safety in terms of growth of the child.
Suggest: : and safety in terms of growth and development of the child.
Line 24: alternative for family interested by plant-based feeding.
Suggest: alternative for families interested in plant-based feeding
Line 28: Cow's milk cannot be replaced, at least during infancy, by milk from sheep, goats, and buffalo due to a high cross reactivity between milk proteins from mammals.
Suggest: In infants with CMPA, cow's milk cannot be replaced by milk from sheep, goats, and buffalo due to a high cross reactivity between milk proteins from mammals.
Line 60: .. but without any reduced….
Suggest: : but without any reduction in intake
Line 80: Children with CMA were at risk of consuming less than the recommended amount of calcium, 15 of 26 vs 12 of 39 in controls (p < 0.05),
Suggest: : Children with CMA were at risk of consuming less than the recommended amount of calcium, 15 of 26 (57.6%) vs 12 of 39 (30.7%) in controls (p < 0.05), i.e I suggest adding % because denominator is different.
Line 186: Nowadays, rice proteins are reconsidered by manufacturers according to valuable properties,
Suggest: Nowadays, rice proteins are considered valuable properties by manufacturers…
Line 253: These indications are demonstrated by generally accepted scientific evidence and their efficacy and safety in young children must be demonstrated by high-quality scientific studies [4].
Suggest ‘generated’ instead of ‘demonstrated’
Line: 285 clinical trial has not been commercialized zed,… Suggest: Omit ‘zed’
Line 279: Unlike CMP eHF, the HRPF as here before mentioned above are not reimbursed by French health insurance.
Suggest: Omit full stop after eHF. Consider: ..as previously mentioned above….
Line 323:The Study of Reche et al. (Blemil Arroz® ) [68], showed and identical infant rejection in HRPF and CMP-eHF formulas (i.e.2 /46 infants against HRPF and 2 / 46 infants against for - CMP-eHF).
Suggest: The Study of Reche et al. (Blemil Arroz® ) [68], showed and identical infant rejection in HRPF and CMP-eHF formulas (i.e.2 /46 infants against HRPF and 2/46 infants against CMP-eHF).
Line 235:.. at pH 2 to 8 while their intact counterpart rice endosperm protein concentrate displayed very bad solubility values [52]. Consider ‘low’ instead of ‘very bad’.
Line 327: ..to a drop-out in 3 of 40 infants as included.
Suggest: …to a drop-out in 3 of 40 included infants.
Line 459: CMP-eHF accounts for 48.4% of the total FSMP used for CMPA in France, HRPF 39.3% and free amino acid preparations 12.3%. (insert full stop here)
Line 463: They are found in a growing number of regions such as North Africa, the Middle East and South America, still not available in many countries in Europe, the United States, Canada, Australia and New Zealand.
Suggest: They are found in a growing number of regions such as North Africa, the Middle East and South America, but are still not available in many countries in Europe, the United States, Canada, Australia and New Zealand.
Line 479: The effect of HRPF on the duration of the CMPA, significantly compared to the effect of CMP-eHF , remains insufficiently documented.
Suggest: The effect of HRPF on the duration of the CMPA, in particular compared to the effect of CMP-eHF , remains insufficiently documented.
Line 482: The quality of rice proteins and the legal frame of FSMPs allows feeding infants and children with no risk of nutritional deficiency.
Please clarify this sentence. Do you mean? The legal framework permits the feeding of FSMPs to infants and children, without any risk of nutritional deficiency, due to the quality of rice proteins.
Author Response
Dear Mrs., Mr.
First, I would like to thank you for correcting our paper and for your accurate advices that they’ve been taken very seriously.
All your suggestions have been considered and we’ve made all the modifications you suggested.
Kind regards,
Round 2
Reviewer 1 Report
The manuscript has been slightly improved, and it is more readable. The manuscript covers an interesting and trendy topic but part of the text is not adding anything new to the existing literature. This is particularly the case in sections 4 & 5. As commented in my previous letter, the text is really similar or even identical to what you already published previously (Bocquet, A., Dupont, C., Chouraqui, J. P., Darmaun, D., Feillet, F., Frelut, M. L., ... & Simeoni, U. (2019). Efficacy and safety of hydrolyzed rice-protein formulas for the treatment of cow's milk protein allergy. Archives de Pédiatrie, 26(4), 238-246.). Therefore, I strongly suggest that sections 4 & 5 should be fully rewritten. It is imperative that you do not describe the same things you already mention in your published work. Per your convenience, the following table (pls see document attached) evidences the similarities between your published work and your article under review in Nutrients. As such, I just do not see how you are making a significant contribution to the literature.
Kind Regards

Author Response
Dear Mrs., Mr.
Thank you for your help in the improvement of the paper.
The manuscript covers an interesting and trendy topic but part of the text is not adding anything new to the existing literature. This is particularly the case in sections 4 & 5
If, as underlined, section 4 and 5 are not new topics, we are convinced that they are essential to understand the entire paper. Therefore, we summarized them into 4 different tables, to give the reader the main information.
We hope this version will fit.
Sincerely
C.Dupont and al.
Round 3
Reviewer 1 Report
Dear Authors,
Thank you for taking into consideration the proposed changes. The manuscript has been improved and now is contributing to the literature by providing a relevant summary of existing knowledge.